# INTERPRETABLE BRAIN-INSPIRED REPRESENTATIONS IMPROVE RL PERFORMANCE ON VISUAL NAVIGATION TASKS

## ABSTRACT

Visual navigation requires a wide range of capabilities in an agent. A crucial one is the ability to determine the agent's own location and heading in an environment. However, existing navigation approaches either assume this information is given, or use methods that lack a suitable inductive bias and accumulate error over time. Inspired by neuroscience research, the method of slow feature analysis (SFA) overcomes these limitations and extracts agent location and heading from a visual data stream, but has not been combined with modern, deep reinforcement learning agents. In this paper, we compare SFA representations with those learned by convolutional neural networks in deep RL agents. We also demonstrate how using SFA representations can improve navigation performance. Lastly, we empirically and conceptually investigate the limitations of SFA and discuss how they currently prevent it from being used more widely for visual navigation in RL.

## 1 INTRODUCTION

Visual navigation is a complex and increasingly relevant task in robotics and in machine learning (ML). Research in this field touches on a wide range of agent capabilities, including the parsing of tasks (Wang et al., 2021), locating objects to interact with (Lyu et al., 2022), mapping out the environment (Chaplot et al., 2020) and planning (Gupta et al., 2017). A basic necessity in navigation, however, is that the agent has to find and move along a path to its target. Finding a path to some location, crucially, requires awareness of one's own location and heading. Unsurprisingly, it has been found that an agent's ability of self-localization is important for navigation and especially long-term planning in ML (Zhu et al., 2021).

In computational neuroscience, slow feature analysis (SFA) (Wiskott & Sejnowski, 2002) is a method modeled on the human visual system that has long been known for its ability to extract position and head direction from a visual stream. In fact, the representations it generates have been related to place cells and head-direction cells, among others (Franzius et al., 2007). This paper illustrates the potential of using SFA representations for deep reinforcement learning (RL) agents on visual navigation tasks.

The contributions of this paper are threefold:

- We explain how SFA representations conceptually differ significantly from current approaches to localization for visual navigation in RL. Other methods either require integration of information over time or lack a suitable inductive bias for extracting interpretable location and heading information from images. SFA addresses both weaknesses.

- We show empirically that SFA representations are not only capable of extracting location and heading, they also make navigation more efficient than representations which do not contain such information. In particular, we show that regular convolutional neural networks in deep RL agents do not learn to extract this information.

- We explain limitations which currently prevent SFA from seamless integration into RL agents, in particular a lack of gradient-based training procedures and the requirements on environment coverage in training data.

This paper aims to present SFA as a suitable and underdeveloped representation learning method for visual navigation, while also investigating its current limitations for this purpose.

## 2 RELATED WORK

**Localization for Navigation**   Representation learning in the context of RL and navigation is often approached through auxiliary tasks (Lange et al., 2023; Jaderberg et al., 2017; Ye et al., 2021a; Mirowski et al., 2017), often without explicitly considering position, orientation or pose of an agent. The works that do use these features, however, can broadly be split into three categories.

The first approach is to just assume the agent is provided with ground truth information on its current absolute location and heading (Ye et al., 2021a;b). The second approach can be called location through integration. It assumes that current changes in position and direction can either be inferred or are provided to the agent. These are then integrated over time (Mirowski et al., 2017). Simultaneous localization and mapping (SLAM) methods are a particularly prominent algorithm class that relies on this approach (Chaplot et al., 2020; Campos et al., 2021). The third approach employs neural networks (commonly convolutional neural networks (CNNs) combined with recurrent neural networks (RNNs)) to learn representations from visual input (Mousavian et al., 2019). These networks do not have any inductive bias towards learning position or heading in particular, although they might be trained in a supervised way directly on this information (Wang et al., 2017; Datta et al., 2021). Both these papers, additionally, still implicitly integrate changes in location. As a consequence, they share the main weakness of the second approach: accumulation of errors over time, which is nicely demonstrated in Figures 4, 6 and 8 of (Wang et al., 2017).

**Navigation with Slow Feature Analysis**   First introduced by Wiskott (Wiskott, 1998; Wiskott & Sejnowski, 2002), SFA was extended to hierarchical networks – not unsimilar to CNNs – in Wiskott (2000). Franzius et al. Franzius et al. (2007) show how hierarchical SFA (hSFA) can be used with independent component analysis (ICA) to extract location and head direction (resembling the neuroscientific concepts of place cells and head direction cells) in a neurologically plausible way from the visual input stream of a simulated animal. Beyond first-person visual input, and potentially also interesting for navigation, the same authors have also used hSFA for object recognition (Franzius et al., 2011). Based on this work, Legenstein et al. (2010) have first applied hSFA to RL: Using hSFA-generated representations, they learn a simple Q-function to make a fish in a tank, seen from above, move to a target. The most recent inspiration for this paper, finally, comes from Schönfeld & Wiskott (2013) who have designed a virtual maze for a virtual rat and use this to extract location and head direction from visual input. Since 2013, the fields of deep learning, reinforcement learning and visual navigation have come a long way. Yet, to the best of our knowledge, there have been no works on visual navigation with hSFA representations since. A gradient-based approach to SFA has been used in the context of RL (Hakenes & Glasmachers, 2019), but not for navigation.

## 3 LEARNING SLOW FEATURES

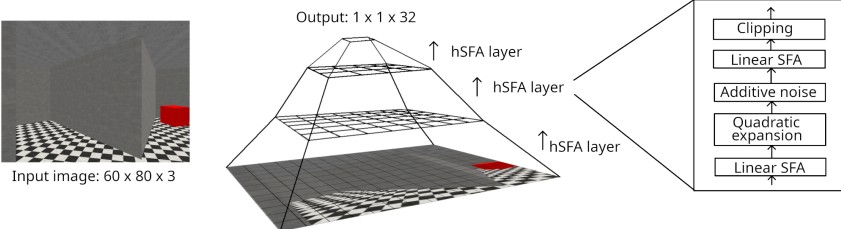

Figure 1: Illustration of the architecture of a hierarchical slow feature analysis model. The input image is perceived in patches by receptive fields with certain strides. These patches are stacked and passed as batches through an hSFA layer.

**Slow Feature Analysis**   Slow feature analysis is based on the slowness principle: Invariant or slowly varying features in a signal are usually of more interest than quickly varying features, which

are often closer to noise. In a visual stream, for instance, individual pixels will vary very quickly while objects or an agent's position do not. To extract slow features from a signal, SFA solves the following optimization problem: Given a (commonly multidimensional) signal $x(t)$, find mappings $y_j(t) = g_j(x(t))$ such that

$$\Delta y_j := \langle \dot{y}_j^2 \rangle_t \tag{1}$$

is minimized under the constraints

$$\langle y_j \rangle_t = 0 \quad \text{(zero mean)} \tag{2}$$

$$\langle y_j^2 \rangle_t = 1 \quad \text{(unit variance)} \tag{3}$$

$$\forall i < j : \langle y_i y_j \rangle_t = 0 \quad \text{(decorrelation and order)} . \tag{4}$$

Here $\langle \cdot \rangle_t$ denotes the temporal mean and $\dot{y}$ the temporal derivative of $y$. The extracted signals $y_j(t)$ are the slowest ones which can be created from $x(t)$ given a family of mapping functions $\mathcal{G}$. The constraints guarantee that trivial solutions (a constant signal) are excluded and that output signals are decorrelated and ordered by slowness. For linear SFA, $g_j \in \mathcal{G}$ are chosen to be linear.

In practice, this results in the following algorithm: First, the signal is whitened to obtain zero mean and identity covariance. As an approximation of the temporal derivative, subsequent data points in the time series signal are then subtracted from each other. Lastly, principal component analysis (PCA) is performed on the differentiated time series. The resulting linear components are already decorrelated and ordered by variance. Since their variance now corresponds to the temporal variance in original data, components are ordered by lowest rather than highest variance.

**Non-linear SFA** The family of linear functions is limited in their ability to extract interesting information. Therefore, non-linear expansion – commonly quadratic expansion – is used on the input signal before performing SFA. hSFA also uses this expansion as opposed to other non-linearities, despite its downside of significantly expanding data dimensionality before processing.

**Hierarchical SFA** In order to deal with visual input streams, or videos, hSFA stacks layers of non-linear SFA modules on top of each other (see Figure 1). One such layer consists of five components: A linear SFA step first reduces the dimensionality of the data. A quadratic expansion then introduces non-linearity and Gaussian noise is added (during training only) to increase training stability. Finally, another linear SFA extracts the slow features. These features are then clipped, commonly and also in this paper to $[-4, 4]$, to avoid propagation of extreme values. Altogether, this whole hSFA layer is commonly referred to as a step of quadratic SFA.

Each but the top-most layer operates on receptive fields with certain strides, similar to a CNN. Moving a receptive field across the image creates image patches. These patches are flattened and treated as batches to train a hSFA layer, similar to weight sharing in a CNN. The top-most layer in hSFA is always a quadratic SFA layer that just works on the flattened output of the second-to-top layer. This is comparable to a linear layer at the end of a CNN, it flattens the output and finally allows all parts of the image to have an effect on any dimension of the output.

In contrast to neural networks, the layers of hSFA, at their core, contain singular value decompositions. The system is therefore trained layer by layer, instead of end-to-end with gradient descent like an artificial neural network. Additional control of extracted features can be obtained by using independent component analysis or learning rate adaptation, which are discussed in the Appendix.

## 4 EXPERIMENTS

This section presents the RL environments and agents that we use for our investigation. It also describes the training of the hSFA, PCA and CNN feature extractors.

### 4.1 ENVIRONMENTS

We use 3D visual navigation environments of the Miniworld package (Chevalier-Boisvert et al., 2023). Each environment contains one red cube representing the target. The task is always to reach the target. There are no other objects present. Observations are $60 \times 80$ pixel RGB images which show the current front view of the agent in the simulated world. There are three possible actions

available: 1) Turn left by $\pi/12$ radians; 2) Turn right by $\pi/12$ radians; 3) Move a small, fixed step forward. We evaluate performance in terms of episode length $l$ rather than reward $r$. Episode length is a more interpretable measure and contains the same information as the reward, which is calculated as $r = 1 - 0.2\frac{l}{l_{\max}}$. A reward is only made available to the agent once it reaches the box.

Exemplary observations of each environment are shown in Figure 6 in the Appendix. Top-views of their layouts are shown when SFA representations are presented in Figure 2. Some of the listed environments are customized, their code is available online (Anonymous, 2025b).

**StarMazeArm**   The target in StarMazeArm is always at the end of the same arm. The initial agent position is a random location in the center room of the maze, its initial heading is random. Maximum episode length is 1500. The optimal policy is to turn until facing the target and then walk forward. In theory this does not require locating the target, as it is always in the same place.

**StarMazeRandom**   This environment is identical to StarMazeArm with the exception that the target is placed in a completely random position each episode. The optimal policy is the same as with StarMazeArm. As opposed to StarMazeArm, however, the agent first has to locate the target in each episode before it can know where to walk.

**WallGap**   The initial agent position is always in the upper room, the initial target position in the lower room. Initial agent heading is random. Maximum episode length is 300. As opposed to the other environments, both rooms have the same textures and thus look visually identical apart from one distant skyscraper This introduces visual symmetries that often make it impossible to extract position and heading from one image alone. The best policy is to walk straight to the gap between rooms, turn to face the target and walk straight to it.

**FourColoredRooms**   The initial agent position and heading are random, as is the target position. Maximum episode length is 250. As opposed to the previous three environments, the wall textures are unique for each wall. Each of the four rooms has a different color, similar to Prince Prospero's rooms in Edgar Allan Poe's The Masque of the Red Death (Poe, 1842). Each wall in a room has a different brightness so that, in contrast to WallGap, there are no visual symmetries despite the symmetry of the layout. The main difficulty is that the number of different rooms makes an exploration strategy necessary to traverse rooms in search of the target.

### 4.2   RL AGENTS

We train PPO agents with different feature extractors (described below) to solve each navigation task. PPO is a simple, general, state-of-the-art, on-policy, model-free policy optimization algorithm in RL (Schulman et al., 2017). Simple here means that it does not involve any navigation capabilities stated in Related work, such as mapping or planning. We use the implementation of Stable Baselines3 (Raffin et al., 2021) to train five agents with random seeds per setup. Details and hyperparameters can be found in the Appendix. We made all code required to reproduce our experiments and results available on GitHub (Anonymous, 2025a)[1].

In addition to agents trained with feature extractors, we report performance of an agent following random performance and an agent following an optimal policy for comparison. The first quantifies the average episode length achieved by 100 random agents on each environment. The second quantifies average episode length of 10 manual runs per environment when following the optimal strategy and exploiting a top-view that includes both agent and target and is not part of the observation.

### 4.3   FEATURE EXTRACTORS

We use a hSFA feature extractor, two CNNs and a PCA feature extractor with PPO. They are described below, more details can be found in the Appendix (Table 3).

**hSFA**   The hSFA feature extractor is pre-trained individually for each environment layout, i.e. only once for StarMaze. We use the sklearn-sfa implementation by Schüler & Lange (2023) to extract

---

[1] https://anonymous.4open.science/r/sfa-for-navigation-6222

representations with 32 features. The pretraining is done on 80,000 data points collected by an agent following a random policy. While 80,000 is a high number, such an amount of data is cheap to collect and in this work we focus on demonstrating the capabilities of hSFA rather than exploring the limits of its hyperparameters. Our experience indicates that far fewer points should be sufficient if collecting them were to be expensive. It is however important that they cover a representative sample of combinations of all locations and headings that the agent might later experience.

Training data is collected on empty environments, i.e. we remove the target cube in order to cover all locations and headings, even those that would otherwise be blocked by the target. The hSFA representations are thus not trained on observations with targets; however the results indicate that the visual cue of a target might still end up being encoded within representations during inference.

Regular resets at maximum episode length of each environment ensure a uniform coverage of the environment, which we found to benefit representations. They do, however, also introduce discontinuities and therefore quick changes in location and heading. We found that these discontinuities do not influence learned representations noticeably if episodes are sufficiently long. While we do not use learning rate adaptation in this work, it could be employed in order to reduce the influence of discontinuities on the representation (see Appendix). During the training of PPO agents, the pre-trained hSFA feature extractor is used to pre-process the observations fed into the PPO algorithm.

**CNNs** We train two CNN architectures to compare hSFA against. They are prepended to the PPO agent and trained jointly with the agent, i.e. on the RL learning task. The first is NatureCNN (Mnih et al., 2015), the default for processing visual observations in Stable Baselines3. Its purpose is to compare hSFA representations with those that do not have an inductive bias towards encoding location and heading. Additionally, we employ a CustomCNN which mimics the architecture of hSFA. This is to show that the advantage of hSFA comes not from its architecture but from its optimization target.

**PCA** Finally, we train a basic PCA feature extractor. Like hSFA it is also pre-trained, on the same data as hSFA, and then used to pre-process observations when training PPO. We use the scikit-learn implementation (Pedregosa et al., 2011). The PCA representations consist of 32 features and explain a surprising cumulative amount of variance: 81.9% for the StarMaze environments, 92.2% for WallGap and 91.0% for FourColoredRooms. The purpose of PCA is to show what PPO itself, without any ability to learn complex features, is able to achieve.

## 5 RESULTS

This section presents and describes the hSFA representations in comparison to those learned by PCA and CNNs, as well as agent performance and behavior.

### 5.1 REPRESENTATIONS

The representations learned by hSFA are analysed on test sets of 80,000 points, sampled for each environment in the same way as the training data for hSFA was sampled. Information in individual hSFA features is visualized by plotting a top view of the agent's positions and coloring each point by the value of a given feature. Images are shown in Figure 2 for the first 6 out of 32 hSFA features. Since the train and test set were sampled in the same way, these images additionally provide an intuition on the environment coverage provided by the train set.

In Figures 7 and 9 of the Appendix we also show the representations learned by PCA and NatureCNN for comparison. These were obtained using the same procedure as with hSFA, only the 512 dimensions of the NatureCNN output were additionally passed through a PCA dimensionality reduction to be able to order and display them.

**Location** Figure 2 shows that location information is encoded in hSFA features. A feature might activate at different locations, for instance feature 1 of FourColoredRooms has a different but constant value inside each room and feature 5 for StarMaze is only positive in the center of the maze. Different components encode different information about location. Earlier components tend to en-

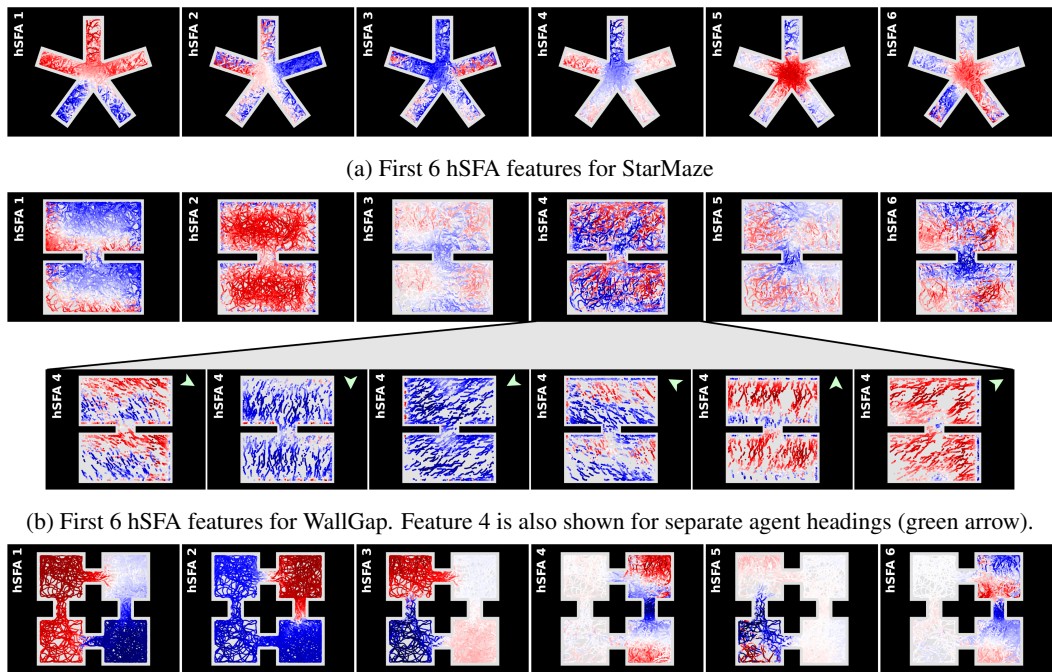

(a) First 6 hSFA features for StarMaze

(b) First 6 hSFA features for WallGap. Feature 4 is also shown for separate agent headings (green arrow).

(c) First 6 hSFA features for FourColoredRooms

Figure 2: Analysis of hSFA representations in different environments (top view). Figures 2a, 2b, 2c show activations of the first 6 hSFA feature dimensions for different positions and orientations in the room. The points are generated by a random agent moving for 80,000 steps without reset. Colors fade from deep red for large positive values into white for zero into deep blue for large negative values. Figure 2b additionally shows the 4th feature of WallGap for separate agent headings.

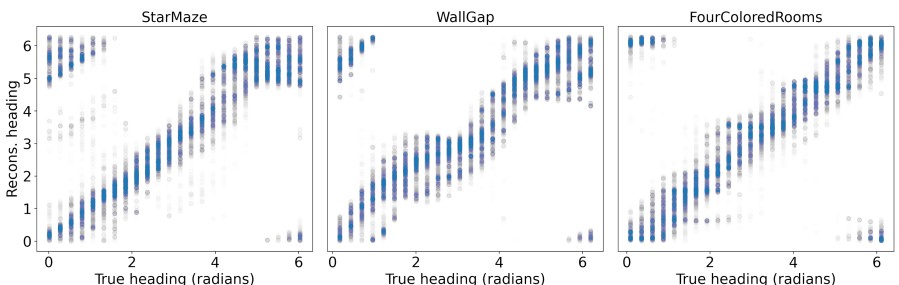

Figure 3: Reconstruction of heading angles. The angle is reconstructed from sine and cosine, which are provided by two linear models trained on all 32 hSFA features. Points are transparent to indicate density. The top left and bottom right corners contain points because of the heading's circularity.

code global information, later components tend to encode local information. The more components are used, the finer the resulting resolution of location can become.

We also find that components are robust: Even in the StarMaze environment, the maze arms can be confidently differentiated although the only visual information that breaks symmetry is the checkerboard pattern of the floor texture being intersected by walls at different angles (see Figure 6 in the Appendix). If, however, observations in different positions look exactly identical, this symmetry cannot be resolved by hSFA. WallGap illustrates this issue in Figure 2b, where representations are the same in each room. In FourColoredRooms this problem does not arise despite the symmetry of its layout, because each wall has a different color.

Both PCA and NatureCNN are also able to resolve some location information. Their representations however are much more limited in their interpretability. A seemingly meaningful representation of

| Agent | StarMazeArm | StarMazeRandom | WallGap | FourColoredRooms |
|-------|-------------|----------------|---------|------------------|
| hSFA | **69** (52, 82) | **147** (92, 227) | 277 (184, 300) | 232 (225, 239) |
| NatureCNN | 415 (270, 592) | 309 (226, 430) | 266 (233, 300) | **187** (178, 201) |
| CustomCNN | 652 (288, 1487) | 364 (194, 443) | 300 (299, 300) | 237 (211, 250) |
| PCA | 773 (621, 1069) | 1005 (911, 1099) | **179** (168, 191) | 222 (212, 233) |
| Random | 1134 (53, 1500) | 1073 (1, 1500) | 300 (300, 300) | 231 (9, 250) |
| Optimal | 36 | 16 | 76 | 53 |

Table 1: Average episode lengths achieved by agents with different feature extractors on the different Miniworld environments, at the end of their training. Minimum and maximum of five agents (100 agents for the random policy) are reported in brackets. Best performance is marked in bold. The reported optimal performance is also an average.

location information is only present in few dimensions and these are considerably noisier than those produced by hSFA.

**Heading**   The heading information encoded in a hSFA feature becomes a pile of intermingled lines of different colors in our visualization. The feature takes different values for different headings, and the lines arise because even a random agent often walks a couple of subsequent steps into the same direction. To illustrate the heading information, we disentangle these lines for the fourth hSFA feature of WallGap, shown in Figure 2b. To do this, we divide a full circle into 6 angle sections (the arrow indicates the center of each angle section). Each image in the lower row only shows the values where the agent's heading falls into a given angle section. The illustration shows that feature values are negative when the agent looks south-west, positive when it looks north-east, and undergo a transition phase between these.

In the case of PCA, this same kind of pattern for heading information is also present (see Figure 7b in the Appendix), although it is less obvious. In the NatureCNN representations, however, Figure 9b in the Appendix shows that the noise does not in any way seem to encode heading. For hSFA, heading generally tends to be encoded in later features (for an explanation, see the ICA and LRA section in the Appendix or (Franzius et al., 2007)). Heading information is thus not visible in many of the images of Figure 2. Still, the angle can be reconstructed well from the first 32 hSFA features, as Figure 3 shows. This reconstruction, based on linear regression, is accurate to within a few degrees.

To reconstruct heading as in Figure 3 and Figures 8 and 10 of the Appendix, we learn two linear regressions that map the features to $\sin(\varphi - \pi)$ and to $\cos(\varphi - \pi)$, where $\varphi$ is the heading. Its value is then reconstructed from the sine and cosine values. Regressions are trained on the first 40,000 steps of our test data and evaluated on the remaining 40,000 steps. It is necessary to use sine and cosine here because heading is a circular variable with a discontinuity from $2\pi$ to $0$. Circular variables (not only angles) are always encoded by their sine and cosine by hSFA, as these do not contain discontinuities and thus vary slowly. The circular nature of the heading is also directly visible in the transition phases for different headings in Figure 2b. Despite PCA not relying on slowness, the same reconstruction technique leads to similar results regarding heading for PCA (Figure 8), even though it effectively returns noise for NatureCNN representations (Figure 10).

## 5.2   RL AGENTS

The average episode lengths achieved by trained agents are reported in Table 1, the performance throughout training is reported in Figure 4 in the Appendix. hSFA agents are more successful than other agents on the StarMaze environments, but not on WallGap or FourColoredRooms. In the latter two, no agent performs close to optimal, although the PCA agent outperforms others on WallGap and the NatureCNN agent has a lead in FourColoredRooms.

In the following, we describe observed behavior for the best out of five agents in each setup. These observations provide a deeper insight than the values in Table 1. Since CustomCNN is consistently outperformed by NatureCNN, we only investigate the more successful behavior of NatureCNN. Additionally, we analyze the performance of baseline agents as well as agents trained on combinations of representations.

**StarMazeArm**    The hSFA agent immediately turns in the right direction towards the target and then walks straight to it, even if the target is not immediately visible. Non-optimal performance is explained by the agent sometimes getting stuck at a protruding corner, which is something that regularly happens to all agents across all environments. The PCA agent walks to the target when it is visible, otherwise it wanders into a random direction until it gets stuck in a wall. The NatureCNN agent also walks to the target when visible and walks in circles otherwise.

**StarMazeRandom**    The hSFA agent walks in circles around the center room until it sees the target, then walks straight to it. In contrast to this, the NatureCNN agent only spins around itself until it sees the target. If it spawned in a location from which it cannot see the target, it spins until the episode ends. The PCA agent displays the same behavior as in StarMazeArm.

**WallGap**    The hSFA agent sometimes manages to walk directly to the gap connecting both rooms, but it often seems confused about the correct direction and ends up walking the wrong direction. If it makes it to the gap, it spins around until it sees the target. If it sees the target it walks towards it. In most cases, it never reaches this last step. The NatureCNN agent walks around randomly until it happens to see the target and then walks straight to it. In many cases the episode ends before it found the target. The PCA agent walks around randomly until it sees the gap. Then it walks straight to the gap. Then it wanders randomly until it sees the cube and walks straight to the cube.

**FourColoredRooms**    The hSFA agent walks around almost randomly, often making some distance and covering most of the room it is in. It makes no effort to search for the target in other rooms. If the target becomes visible, the agent does not react to it. Instead the agent seems to rely on hitting the target by walking around. The NatureCNN agent also walks around randomly, however it turns more and covers significantly smaller distances. While it does not actively seem to search for the target, it does walk straight to the target when it becomes visible. The PCA agent walks around randomly and covers as much ground as the hSFA agent, albeit in a seemingly more random way. It does however tend to walk to gaps between rooms or to the target when either becomes visible. It does so in a less directed and straightforward way than the NatureCNN agent so that the episode often ends before the target is reached.

**Baselines**    In addition to the primary hSFA, CNN and PCA feature extractors we introduce some baselines for comparison and report these results in Figure 5 in the Appendix. The ground truth baseline provides location and heading $\varphi$ (in the form of $\cos(\varphi)$ and $\sin(\varphi)$) directly from the environment to the agent. The integrated noisy baseline cumulatively adds Gaussian noise to the ground truth at each step of an episode, simulating related work that integrates changes in location over time. The noise has a variance of 1/250th of the value range of the respective variables, which is calibrated to approximately match the deviations in Wang et al. (2017). Because both these baselines are unable to visually locate the randomly placed goal in the other environments, they can only work well in StarMazeArm. Here however, both baselines interestingly perform worse than hSFA – likely because the hSFA representations structure location and heading information in a way that is easier to process for an agent: They calculate more features, each of which encodes the information at a different level of coarseness. In comparison to the pure ground truth and also to hSFA, the integrated noisy representation is less reliable and can vary strongly from run to run. This is indicated by the shaded areas in Figure 5 that illustrate best and worst agent performance.

**Combined representations**    Finally, Figure 5 in the Appendix also presents agents that are trained on combinations of visual information as well as location and heading. Visual information is processed by the NatureCNN and combined with either the location and heading ground truth or with the learned hSFA representations. In WallGap and FourColoredRooms, these baselines fail to learn reliably. In WallGap this is likely because the narrow gap prevents reaching the goal often enough that the agent does not get sufficient learning feedback. In FourColoredRooms, these agents still lack the memory required for targeted exploration. Interestingly, the combinations of representations are not able to outperform hSFA in the StarMaze environments. As with the baselines, this points to a favorable structure of hSFA representations. In addition, the agents with combined representations experience a decrease in performance after a certain amount of training on StarMazeRandom, which happens later for hSFA representations than for ground truth representations. The agents using hSFA representations also perform better than those with location and heading ground truth.

## 6 DISCUSSION & CONCLUSION

This section discusses how the hSFA representations compare to CNN and PCA representations and how useful they are for visual navigation. It also states current limitations of using hSFA for navigation in RL.

**Representations** Our results show that hSFA is consistently able to extract information about both location and heading of the agent, unless there are visual symmetries as in WallGap. We stress again that these symmetries are unlikely outside simple simulated environments. It is important to note that because the hSFA algorithm directly calculates the solution to a mathematical optimization problem, it is imperative that its outputs are the slowest signals that can be extracted from the input, given the function family $\mathcal{G}$ that its architecture is restricted to. This stands in contrast to neural networks, where the quality of representations often depends on random seed and initialization (Locatello et al., 2019). Furthermore, it implies that if hSFA representations do not encode location and heading, then these are not the slowest signals contained in the visual input stream. Such a thing can happen for various reasons, for instance due to boundary conditions of an environment that result in discontinuities in heading or location. We conclude that the slowness principle is a valid and powerful inductive bias for extracting location and heading in the visual input stream of an agent. Furthermore, the hSFA algorithm is a suitable architecture to obtain such representations and thus obtains more interpretable representations than those produced by PCA or a CNN. A comparison to passing ground truth location and heading into the agent additionally suggests that hSFA presents location and heading information in a particularly accessible way, likely because its individual decorrelated features contain different granularity levels of location and heading, based on their respective slowness from the perspective of the moving agent.

**Navigation with hSFA** The StarMazeArm environment shows that hSFA representations with an explicit and interpretable encoding of location and heading make visual navigation simpler for an RL agent. The agent's movements across all environments except WallGap, in fact, become more purposeful and confident due to the agent's increased awareness of its presence in relation to its surroundings. In addition, the success on StarMazeRandom illustrates that hSFA representations are able to retain information about the visual scene – in this case whether the target cube is visible – in addition to location and heading. The fact that the target cube is ignored by the hSFA agent in FourColoredRooms, on the other hand, suggests that extraction of visual cues with current implementations of hSFA has its limit. It has however been shown before that positions of slowly moving objects can be extracted if hSFA is trained on such data (Franzius et al., 2008). The question is how slow these features are compared to agent location and heading, and thus how many hSFA features are required to obtain this information. Although this work does not explicitly examine robustness to environmental perturbations such as lighting changes, we refer the reader to Berkes & Wiskott (2005), who demonstrate that hSFA's lower layers learn generally robust, generic visual features and to Schönfeld & Wiskott (2015), who study the effect of environmental layout changes such as morphing and scaling on hSFA performance and find it to be relatively robust to such changes, too.

PCA and CNN are both better at extracting visual cues from an image, as proven by the fact that agents using them walk straight to the target, and sometimes towards gaps, as soon as these become visible. On the other hand, PCA and CNN representations do not or barely encode location and heading significantly worse if at all, as indicated by their comparatively bad performance and inefficient behavior in StarMazeArm. This is true even for CustomCNN, which supports our claim that the slowness bias rather than the particular architecture of an hSFA feature extractor is responsible for learning location and heading.

In general, the results show that PPO is able to solve simple navigation problems when given sufficient representations. The necessity of such representations is illustrated by the comparatively bad performance on simple StarMaze environments with PCA representations, which only compress images into the same dimensionality provided by hSFA. For more complex tasks and environments, such as FourColoredRooms, navigation with a simple RL agent reaches its limits even with meaningful representations. Here, additional capabilities such as planning or mapping become important.

**Limitations** The main limitation of hSFA is that it always only considers the current observation without any context. It shares this limitation with all other approaches we consider in this work.

Likewise, PPO only learns a direct policy and has no inherent planning capacity. The combination of representation learning and planning approaches however has the potential to improve performance also on environments with symmetries (such as WallGap) or partial observability (such as FourColoredRooms).

A disadvantage of hSFA compared to neural networks is its training procedure. Layer-wise training is slow because it is not as optimized as gradient descent. Additionally, its quadratic expansion slows hSFA further. Training an RL agent with hSFA feature extractor is about half as fast as training an agent with prepended CNN feature extractor. A gradient-based SFA version with the potential to address both issues has been proposed by Schüler et al. (2019).

Finally, the data used to train hSFA has to provide a reasonable coverage of positions and headings in an environment. While this is usually easy to obtain by moving around in a random manner for a short while, the algorithm could be improved by introducing the ability of online learning, during exploration of a new area. This is currently not possible.

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

## A APPENDIX

### A.1 AGENT TRAINING

To train our agents, we use the PPO implementation provided by Stable Baselines3 (Raffin et al., 2021). Parameters which are different from default are reported in Table 2. In addition to the reported parameters, we use the CnnPolicy for the NatureCNN feature extractor and the MlpPolicy for the other ones. We train for 1,000,000 steps on the StarMazeArm environment and for 2,000,000 steps on the remaining environments.

The layer specifications of the hSFA, NatureCNN and CustomCNN feature extractors are listed in Table 3. The number of learnable parameters for hSFA, NatureCNN, CustomCNN and PCA are listed in Table 4. hSFA and PCA are pre-trained, and when training the PPO agent they are only used to transform observations into a 32-dimensional feature space. NatureCNN and CustomCNN are trained end-to-end together with the PPO agent on the RL task. Representations by hSFA, PCA and CustomCNN are fed through two fully connected layers of a neural network, as is common for PPO in Stable Baselines3. These layers are identical for all three, trained together with the PPO agent on the RL task, and are listed in Table 3. NatureCNN, on the other hand, is already an internal feature extractor of Stable Baselines3 and its representations are directly used for policy learning. In addition to the results in Table 1 of the main paper, we show the training curves for all agents on all environments in Figures 4 and 5. The combined representations are obtained as follows: The ground truth or hSFA representation, respectively, are expanded by a single linear layer to match the output dimensionality of the NatureCNN output (512). Then both vectors are concatenated and passed into PPO's policy network.

| Parameter | Value |
|---|---|
| n_steps | 128 |
| learning_rate | 0.00025 |
| ent_coef | 0.01 |
| clip_range | 0.1 |
| batch_size | 128 |

Table 2: Parameters used with the PPO model of Stable Baselines3. Only parameters that were not left at their default setting are listed.

| Layer | Type | Receptive field | Stride | Exp. deg. | # Channels out |
|---|---|---|---|---|---|
| hSFA layer 1 | Quadratic SFA | (10, 10) | (5, 5) | 2 | 32 |
| hSFA layer 2 | Quadratic SFA | (3, 3) | (2, 3) | 2 | 32 |
| hSFA layer 3 | Quadratic SFA (fully connected) | – | – | 2 | 32 |
| hSFA MLP 1 | Fully connected | – | – | – | 64 |
| hSFA MLP 2 | Fully connected | – | – | – | 64 |
| PCA MLP 1 | Fully connected | – | – | – | 64 |
| PCA MLP 2 | Fully connected | – | – | – | 64 |
| NatureCNN layer 1 | Convolution | (8, 8) | (4, 4) | – | 32 |
| NatureCNN layer 2 | Convolution | (4, 4) | (2, 2) | – | 64 |
| NatureCNN layer 3 | Convolution | (3, 3) | (1, 1) | – | 64 |
| NatureCNN layer 4 | Fully connected | – | – | – | 512 |
| CustomCNN layer 1 | Convolution | (10, 10) | (5, 5) | – | 32 |
| CustomCNN layer 2 | Convolution | (3, 3) | (2, 2) | – | 32 |
| CustomCNN layer 3 | Convolution | (1, 1) | (1, 1) | – | 32 |
| CustomCNN MLP 1 | Fully connected | – | – | – | 64 |
| CustomCNN MLP 2 | Fully connected | – | – | – | 64 |

Table 3: Parameters used for the hSFA and CNN networks. The MLP layers used with hSFA, PCA and CustomCNN are those introduced by the MlpPolicy in Stable Baselines3. They are automatically appended to the hSFA and CustomCNN feature extractors. In the sklearn-sfa package (Schüler & Lange, 2023), the hSFA layer 3 does not have to be specified. Exp. deg. refers to the degree of expansion, a parameter used in hSFA layers.

| Feature Extractor | # Parameters |
|---|---|
| hSFA | 101,600 |
| PCA | 481,472 |
| NatureCNN | 862,880 |
| CustomCNN | 26,208 |

Table 4: Number of learnable parameters of the different feature extractors.

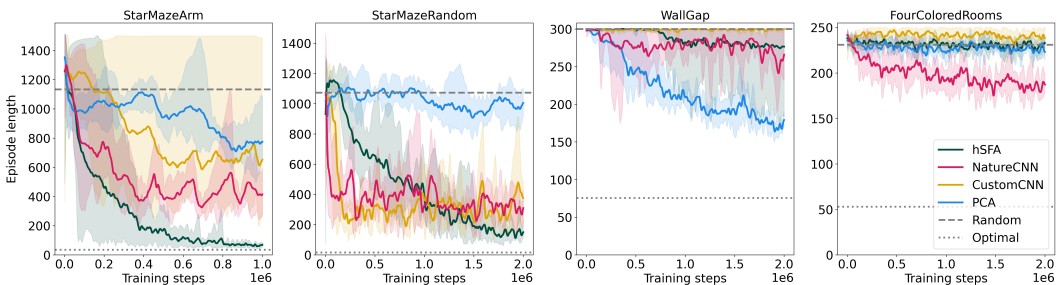

Figure 4: Performance of agents with various feature extractors on the different Miniworld environments. Shaded areas indicate the minimum and maximum of five agents trained with different random seeds. Curves have been smoothed slightly for clearer presentation.

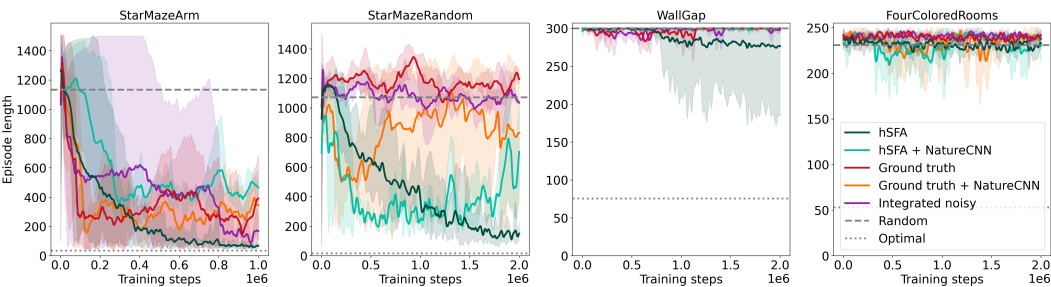

Figure 5: Performance of agents with baseline features (ground truth and integrated noisy, which is ground truth with noise that accumulates over episodes), as well as combined feature extractors, on the different Miniworld environments. Shaded areas indicate the minimum and maximum of three agents trained with different random seeds. Curves have been smoothed slightly for clearer presentation.

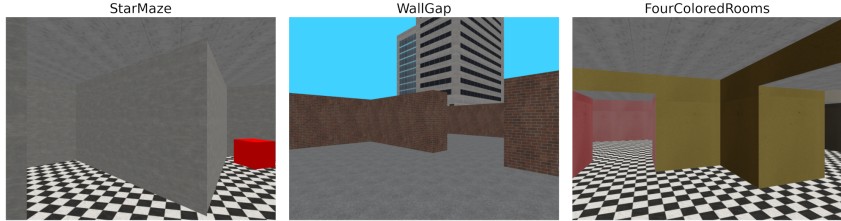

Figure 6: Exemplary observations rendered from the different environments. In this observation of StarMaze, a red target cube is visible. There is no shade from illumination, so the different wall texture colors in FourColoredRooms are in fact textures of different colors.

## A.2 ICA AND LRA

In addition to the base hSFA algorithm, various extensions have been proposed (Escalante-B. & Wiskott, 2012). Two of these can be used to affect the kind of representations extracted from the input. Both have been used to extract location and heading.

The first is a final layer of sparse coding achieved through independent component analysis (ICA), which is attached to the top of the hSFA model by (Franzius et al., 2007; Schönfeld & Wiskott, 2013). The reason for its use is that only this step of sparse coding transforms location information into neuro-plausible place fields. Additionally, (Schönfeld & Wiskott, 2013) claim that the use of ICA was required to disentangle head directions. We find, however, that we can obtain head direction and location information without ICA in this work.

The second is learning rate adaptation (LRA) (Franzius et al., 2007). By weighing data points, their influence on the SFA results can be controlled. In practice, this is achieved by including

weights for the differentiated time series when calculating the covariance matrix that is used for singular value decomposition within SFA (Price, 1972). To calculate weights for the differences between two points, LRA requires an aggregation method. Geometric mean is a good choice, as the arithmetic mean has a tendency to smooth weights out. LRA should be used if there are sudden, fast and unnatural changes in a signal. Two prominent examples are a suddenly mirrored or reflected heading when bouncing into the wall of a simulated environment or an interval of missing data in a time series. These discontinuities would artificially make signals, such as the heading in this example, change faster than they actually do. Such large differences then effectively act similar to how an outlier would in PCA. They strongly affect the whole representation, unless they are mitigated by a small weight. In some cases, it might be practical to apply LRA to certain movements, in particular fast rotations, since rotation of an agent typically changes faster than location when they are normalized by $2\pi$ and size of the environment, respectively. This is proven in (Franzius et al., 2007).

LRA is difficult to use if there are only few discrete actions, as is the case in this work. Larger weights for rotations and smaller weights for moving ahead average each other out for almost all differences of data points collected by a random policy as one action is very often followed by a different one. Additionally, we find that we do not need to use LRA under the conditions examined in this work to obtain good representations.

### A.3 REPRESENTATIONS LEARNED BY PCA AND CNN

Here, we present the representations learned by the PCA feature extractor (Figures 7 for location and 8 for heading) and the NatureCNN feature extractor (Figures 9 for location and 10 for heading). As in the Discussion in the main paper, we omit CustomCNN here due to its inferior performance. Since the dimensionality of the NatureCNN representations is 512, and these have no natural order, we apply a PCA dimensionality reduction to make information contained in the representations more concise and also ordered. For StarMaze, WallGap and FourColoredRooms the first six PCA components capture a cumulative variance ratio of 68%, 89% and 75%, respectively. Since the NatureCNN representations are trained together with the RL algorithm instead of independently, the representations in Figure 9 are those obtained by the best performing RL agent on each environment.

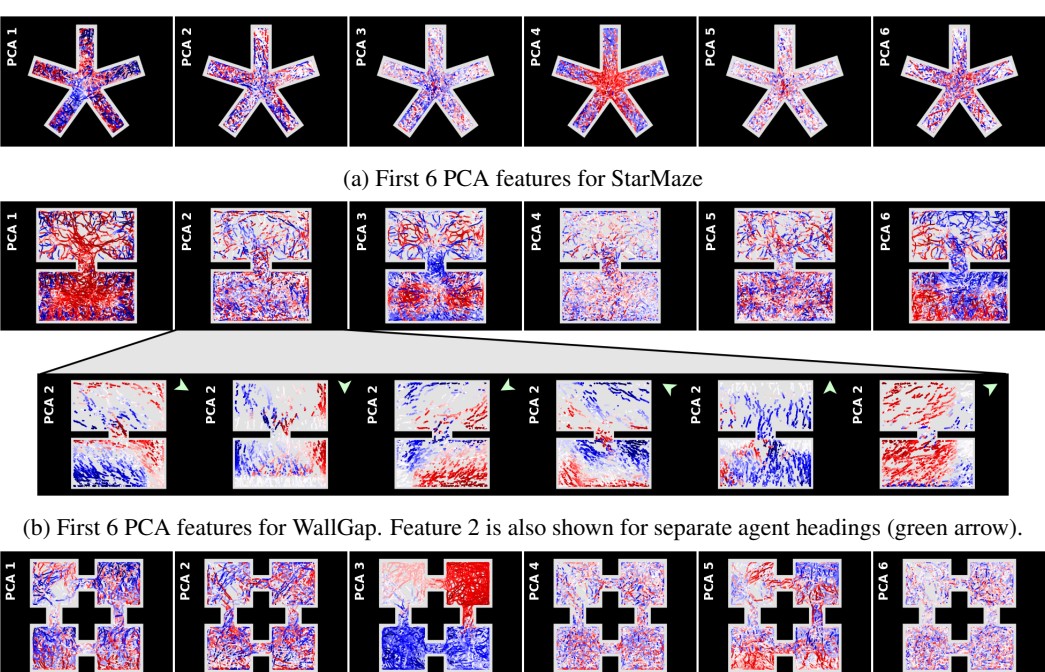

(a) First 6 PCA features for StarMaze

(b) First 6 PCA features for WallGap. Feature 2 is also shown for separate agent headings (green arrow).

(c) First 6 PCA features for FourColoredRooms

Figure 7: Analysis of PCA representations in different environments (top view). Figures 7a, 7b, 7c show activations of the first 6 PCA feature dimensions for different positions and orientations in the room. The points are generated by a random agent moving for 80,000 steps without reset. Colors fade from deep red for large positive values into white for zero into deep blue for large negative values. Figure 7b additionally shows the 2nd feature of WallGap for separate agent headings.

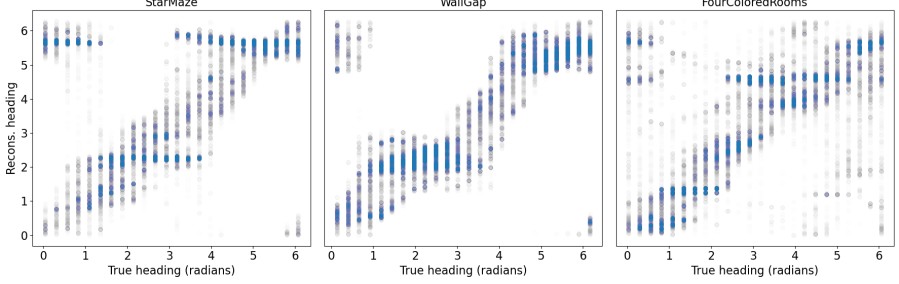

Figure 8: Reconstruction of heading angles for PCA. The angle is reconstructed from sine and cosine, which are provided by two linear models trained on all 32 PCA features. In order to see density, points have a high transparency. The top left and bottom right corners contain points because of the heading's circularity.

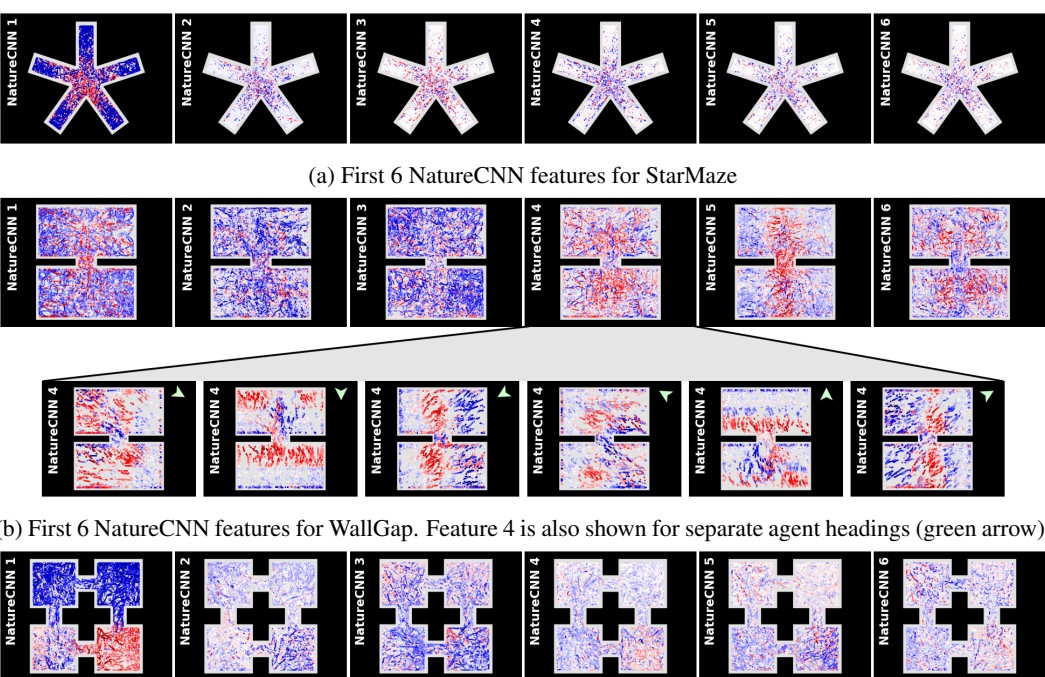

(a) First 6 NatureCNN features for StarMaze

(b) First 6 NatureCNN features for WallGap. Feature 4 is also shown for separate agent headings (green arrow).

(c) First 6 NatureCNN features for FourColoredRooms

Figure 9: Analysis of NatureCNN representations in different environments (top view). The raw unordered 512 NatureCNN features are additionally passed through a PCA dimensionality reduction to obtain more meaningful and ordered visualizations, so these Figures do not show the raw representations returned by NatureCNN. Figures 9a, 9b, 9c show activations of the first 6 PCA components for different positions and orientations in the room. The points are generated by a random agent moving for 80,000 steps without reset. Colors fade from deep red for large positive values into white for zero into deep blue for large negative values. Figure 9b additionally shows the 4th feature of WallGap for separate agent headings.

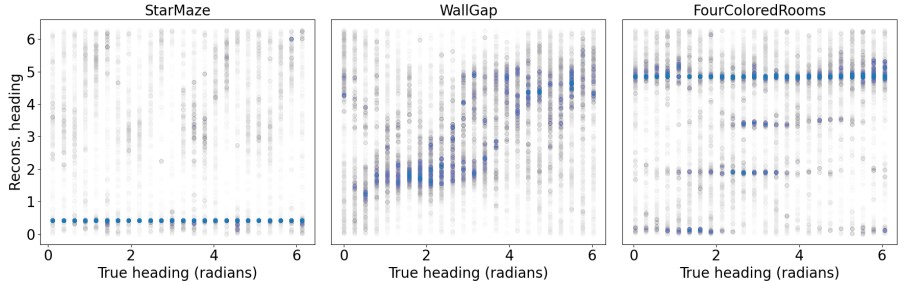

Figure 10: Reconstruction of heading angles for NatureCNN. The angle is reconstructed from sine and cosine, which are provided by two linear models trained on the first 32 dimensions of the output of a PCA dimensionality reduction of the 512-dimensional NatureCNN representations. In order to see density, points have a high transparency.

