# OpenReview forum: "Interpretable Brain-Inspired Representations Improve RL Performance on Visual Navigation Tasks"
_ICLR.cc/2026/Conference — Submitted to ICLR 2026_

### Official Review · Reviewer_WM9V · 2025-10-29

**Soundness:** 3
**Presentation:** 3
**Contribution:** 3
**Rating:** 6
**Confidence:** 4

**Summary:**

The paper investigates whether “interpretable” representations learned by hierarchical Slow Feature Analysis (hSFA) can improve the performance of reinforcement learning agents (PPO) in visual navigation. It compares three representations end to end: pretrained hSFA, representations learned during training by CNNs (NatureCNN and a CustomCNN mirroring hSFA structure), and a PCA baseline. The evaluation spans four MiniWorld environments and reports both interpretability of representations (reconstructing position and heading) and navigation efficiency (average episode length). The main finding is that hSFA outperforms CNNs and PCA on the StarMaze variants, with the first 32 components supporting linear reconstruction of heading, while limitations are discussed such as lack of gradients, coverage requirements, and sensitivity to symmetry.

**Strengths:**

* **Clear motivation and task framing**

  * The paper targets the gap in “self-localization and heading estimation” for visual navigation and surveys limitations of three families of approaches, highlighting hSFA’s inductive bias. This sets a precise theoretical starting point and comparison frame for later design choices.
  * It emphasizes error accumulation in odometry-like integration and uses this to motivate the stability of hSFA.

* **Strong evidence for representation interpretability**

  * Visualizations of position coding show components with distinct global and local spatial sensitivities.
  * Heading can be linearly reconstructed from the first 32 components, with a clear treatment of the circular nature of angles.

* **Insightful analysis of behaviors**

  * On StarMazeArm and StarMazeRandom, hSFA exhibits goal-directed behavior that first orients then proceeds directly, contrasted with CNN and PCA agents that “chase only when visible” or wander.
  * In WallGap, environmental symmetry leads to indistinguishable positions for hSFA and the paper offers a reasonable explanation.
  * In FourColoredRooms, hSFA underperforms CNNs, suggesting CNNs remain strong at extracting certain visual cues.
  * The discussion connects performance differences to the lack of explicit planning or mapping.

* **Openness and reproducibility**

  * Anonymized code links and an environment fork are provided. Training steps, policies, and the SB3 implementation are clearly described.

**Weaknesses:**

* **Baselines and task settings could be stronger**

  * No memory-based visual navigation baseline such as CNN with RNN or LSTM is included to test “integration over time.”
  * There is no perceptual baseline with explicit supervision for heading and position (for example, predicting φ and (x, y) then feeding PPO).
  * Only on-policy PPO is used. There are no off-policy agents or agents augmented with planning or mapping.

* **Limited systematic analysis of coverage and symmetry**

  * Pretraining requires about 80k random trajectories and stresses coverage of position–heading combinations, but there is no degradation curve for undercoverage.
  * The symmetry analysis in WallGap is mostly qualitative and visualization-based, without an information-theoretic lower bound or formal recoverability condition.
  * There is no sensitivity threshold study for “slightly breaking symmetry,” such as adding mild texture noise.
  * There is no report of how different hSFA dimensionalities k affect success rates or errors.

* **Environment-specific pretraining**

  * hSFA must be pretrained separately for each environment and cannot be directly transferred to a new environment. This limits applicability in multi-task or real-world settings.

* **Limited representational generality**

  * The learned dimensions encode position–heading combinations specific to a given environment rather than abstract spatial coordinates or goal-relative positions, which restricts cross-environment policy generalization.

**Questions:**

* **Baselines and task difficulty**

  * Would the authors add a CNN+LSTM or RNN baseline within the same PPO framework, swapping only the feature head and memory module, to test the benefit of temporal integration?
  * Would the authors include an explicitly supervised localization/heading head trained on rendered ground truth, to quantify the trade-off between unsupervised interpretability and supervised performance?
  * Could the authors design occlusion, noise, and lighting perturbation experiments and report robustness curves as performance versus perturbation strength?
  * Can the authors evaluate agents that include mapping or short-horizon planning modules, reusing the same representations, to test gains in more complex settings?

* **Coverage and symmetry analysis**

  * Can the authors report degradation under undercoverage by varying pretraining samples from 10k to 80k and measuring localization error and RL performance, and provide an estimate of minimal usable coverage?
  * Can the authors give an information-theoretic lower bound or formal unidentifiability condition for WallGap and then test the minimal symmetry breaking needed to restore identifiability?

* **Transfer and generalization**

  * Have the authors considered transferring hSFA representations to new environments via joint training, environment-conditioned modulation, or hybridization with more transferable backbones such as CNNs or transformers to improve cross-environment generalization?

---

> ### Author Response · Authors · 2025-11-20
>
> Thank you for the valuable feedback. We have updated and improved the manuscript accordingly, with all changes marked in red in the PDF. In particular, we want to address the following points:
>
> - **Baselines**: To approximate a best-case predictor of heading and location, we trained additional agents that use these environment variables directly as a ground-truth baseline. To approximate temporal integration, we added an integrated noisy baseline that cumulatively adds noise to the ground truth throughout each episode. These results are now included in Figure 5 (Appendix). Both these baselines only work in StarMazeArms, as the other environments require additional visual information to locate the randomly positioned goal. For those environments, we have now also trained and reported results for **combined feature extractors**, which combine ground truth or hSFA features with a CNN. The results of these new agents seem to strengthen our points regarding the advantages of hSFA and are discussed in Section 5.2.
>
> - **Robustness**: We now refer to relevant prior work in Section 6: [1] shows lower-layer hSFA features are quite general and can thus be expected to be robust to perturbations such as lighting, and [2] finds that hSFA representations are relatively robust to certain changes in room layouts (such as morphing and scaling).
>
> - **Planning modules**: We deem an implementation of planning modules, or agent memory, as out of scope for this investigation of representation learning methods. While such combinations of capabilities are a promising direction for future work, here we aim to focus on the representations themselves and to only go as far as illustrating the limits of representation learning with WallGap and FourColoredRooms.
>
> - **Coverage**: We selected 80,000 hSFA samples because visual inspection showed that with fewer random steps, agents sometimes failed to reach parts of the environments. So we selected this deliberately high number to ensure maximally robust hSFA representations. In practice, far fewer samples may suffice, but this strongly depends on environment layout, so we do not think a detailed analysis would be generalizable. The key point is that the agent must have visited approximately all "views" in the environment to form accurate location and heading representations.
>
> - **Transfer**: Jointly training hSFA on several room layouts is an interesting idea but beyond the scope of this work. Since hSFA training requires temporal derivatives of visual data rather than i.i.d. samples, designing a suitable way to combine samples from different environments will require careful consideration and will likely prove critical for success. We leave this for future work.
>
> ----
>
> [1]: Pietro Berkes and Laurenz Wiskott. "Slow feature analysis yields a rich repertoire of complex cell properties." Journal of vision 5.6 (2005): 9-9.
>
> [2]: Fabian Schönfeld and Laurenz Wiskott. "Modeling place field activity with hierarchical slow feature analysis." Frontiers in computational neuroscience 9 (2015): 51.

---

> > ### Comment · Reviewer_WM9V · 2025-11-27
> >
> > Thanks for the authors' response. I will maintain my current recommendation.

---

### Official Review · Reviewer_6pwf · 2025-10-30

**Soundness:** 2
**Presentation:** 3
**Contribution:** 2
**Rating:** 4
**Confidence:** 3

**Summary:**

This article introduces the slow feature analysis (SFA) method to deep reinforcement learning (RL) researchers as a novel feature extraction technique. The authors argue that SFA possesses several advantages for visual navigation tasks, particularly its ability to identify the agent's location and heading. They conducted experiments in four MiniWorld environments and visualized the features extracted by their proposed hSFA method alongside baseline methods. Their visualizations demonstrate that different channels of hSFA are activated at distinct spatial locations, and the extracted features exhibit a notable correlation with the agent's heading. Such patterns are not clearly observed in CNN and PCA feature extractors. The authors also discuss the limitations of SFA to provide insights for future research.

**Strengths:**

The SFA method exhibits clear patterns related to location and heading of the agent. The visualization clearly support the main arguments. Overall, the work is easy to follow.

**Weaknesses:**

The empirical results appear somewhat inconclusive. Specifically, hSFA method outperforms NatureCNN in two environments but underperforms a standard CNN in the other two. Furthermore, the experiments are conducted in environments with relatively simple physics. It remains a question whether SFA alone can scale effectively to environments with more complex physical interactions.

**Questions:**

1. Beyond location and heading, is the features learned by hSFA capable of represent or distinguish specific objects (e.g., walls or obstacles) within the environment?
2. How would the agent's performance change if it were provided with ground-truth location and heading directly? A comparative analysis with such a baseline may yield valuable insights into the empirical results.
3. Given that hSFA and CNNs/PCAs extract complementary types of features, what would be the potential of a hybrid feature extraction architecture for improving navigation performance?

---

> ### Author Response · Authors · 2025-11-20
>
> Thank you for your valuable feedback. It helped us improve the paper; changes are marked in red in the PDF. Responses to your questions, in order:
>
> 1. Although not the focus of this paper, prior work shows that hSFA can recognize and **represent objects**. Specifically, [1, 2] demonstrate the extraction of identity, position, and rotation for single objects, and [3] extends this to multiple objects in an RL environment.
>
> 2. We agree on the missing **ground-truth baseline**. To address this, we trained additional agents using ground-truth features and added results to Figure 5 and Section 5.2. Surprisingly, PPO with ground-truth input performs worse than PPO with hSFA features on StarMazeArm, suggesting that hSFA representations with their normalized, decorrelated features of location and heading information at different granularity levels are easier to process for the agent. hSFA's capability to identify goal-presence from images likely also helps and allows it to perform well even if the goal position is not fixed -- a scenario where the baseline necessarily fails.
>
> 3. An interesting question. To address it, we trained two **hybrid feature extractors** (added to Figure 5 and Section 5.2). Although these learn faster initially, their peak performance is below hSFA alone, again likely due to hSFA’s higher-quality representations. Performance also degrades after a certain amount of training, relatively early for ground truth + CNN and later also for hSFA + CNN. This indicates that the neural networks may unlearn useful features, whereas pretrained and frozen hSFA necessarily remains robust. We believe CNN + hSFA can still be a useful combined extractor in more complex setups, but defer a more thorough investigation of these tradeoffs to future work.
>
> Finally, we dispute that the results are inconclusive: hSFA works well on the StarMaze environments and WallGap and FourColoredRooms were intentionally chosen to illustrate hSFA’s limitations. WallGap highlights its limitations regarding visual symmetry and FourColoredRooms shows that suitable representations alone do not guarantee success in all RL environments.
>
> ----
> [1]: Mathias Franzius, Niko Wilbert, and Laurenz Wiskott. "Invariant object recognition with slow feature analysis." International Conference on Artificial Neural Networks. Berlin, Heidelberg: Springer Berlin Heidelberg, 2008.
>
> [2]: Mathias Franzius, Niko Wilbert, and Laurenz Wiskott. "Invariant object recognition and pose estimation with slow feature analysis." Neural computation 23.9 (2011): 2289-2323.
>
> [3]: Robert Legenstein, Niko Wilbert, and Laurenz Wiskott. "Reinforcement learning on slow features of high-dimensional input streams." PLoS computational biology 6.8 (2010): e1000894.

---

> > ### Comment · Reviewer_6pwf · 2025-11-24
> >
> > Thank you for your comprehensive response. I am not entirely clear on why SFA is able to learn location and heading information. While some relevant references are cited, it would be beneficial to include a direct explanation of the underlying intuition or mechanism in the main text.
> >
> > Given the presented results, the importance of explicit location and heading information appears questionable, as when provided with ground-truth location and heading, the performance is poor. Furthermore, the physics model of environments is relatively simple. The action space is limited to turning left/right by $\frac{\pi}{12}$ radians, and moving a fixed step forward. In light of these factors, the significance of the demonstrated findings is unclear. Therefore, more comprehensive investigation may be needed to establish the substantive contribution of this work.

---

> > > ### Author Response · Authors · 2025-12-03
> > >
> > > Thank you, too, for your response.
> > >
> > > **Why can SFA learn location/heading?**
> > >
> > > As explained in the paper, these features are learned because they change more slowly, from the perspective of a moving agent, than any other features in a static environment. Beyond this, paper [1] explains in detail how SFA is able to extract location and heading. Our paper, instead, seeks to illustrate the potential and the limitations of using such representations for visual spatial navigation with modern deep RL. To keep our paper focused and concise, we decided against adding further information from [1] and point readers towards it in our _Introduction_ and _Related Work_.
> > >
> > > **Relevance of explicit location and heading information**
> > >
> > > We believe that our newly added results show it is the representation of information, rather than only the presence of information, that matters. Intuitively, location and heading are clearly important for navigation. As we argue in our added paragraphs (marked red in the PDF), location and heading alone are not sufficient for most of our presented tasks; a visual input is required to locate the goal. This explains the essentially random performance of using only ground truth location/heading for _StarMazeRandom_, _WallGap_ and _FourColoredRooms_. For _StarMazeArm_ however, explicit location and heading information does work well, but is outperformed by hSFA with its distributed representations. Furthermore, hSFA (with or without CNN) also outperforms "Ground truth + CNN" for _StarMazeRandom_. These are strong empirical indicators that hSFA as a representation of location and heading is (i) important and (ii) superior to ground truth vectors [$x$, $y$, $\cos(\theta)$, $\sin(\theta)$].
> > >
> > > **Simplicity of the physics model**
> > >
> > > We again want to emphasize the scope of our paper, which is the investigation of using hSFA-computed location and heading representations for spatial navigation. We agree that further scenarios with more complex physics and actions, for instance grasping or other kinds of object interactions, are intriguing research directions. However, to keep the focus of this paper on its core message, we defer these to future publications.
> > >
> > > ----
> > > [1]: Mathias Franzius, Henning Sprekeler, and Laurenz Wiskott. Slowness and sparseness lead to place,
> > > head-direction, and spatial-view cells. PLoS computational biology, 3(8):e166, 2007

---

### Official Review · Reviewer_7wYR · 2025-11-01

**Soundness:** 2
**Presentation:** 3
**Contribution:** 2
**Rating:** 2
**Confidence:** 3

**Summary:**

The paper investigates whether hierarchical Slow Feature Analysis (hSFA) can yield interpretable spatial representations—explicit location and heading—that also help RL on visual navigation. The pipeline pretrains hSFA offline from egocentric videos collected with a random policy (target removed), then freezes the extractor in a PPO agent. On Miniworld tasks, hSFA features are linearly decodable for heading and exhibit topographic spatial selectivity; PPO+hSFA outperforms CNN/PCA on localization-centric mazes (StarMaze) but does not dominate on symmetry-heavy or cue-driven tasks (WallGap/FourColoredRooms). The paper discusses why SFA helps (stable/slow spatial codes) and where it struggles (symmetries, offline coverage, lack of end-to-end gradients), and suggests gradient-based SFA and hybrid SFA+memory/mapping as next steps.

**Strengths:**

* **Clear motivation.** Articulates why localization-relevant inductive bias (slowness) could aid navigation versus integration-based odometry or generic CNNs.
* **Interpretability evidence.** Spatial maps and linear probes show human-readable structure for heading/position.
* **Task relevance.** Benefits materialize where self-localization is the bottleneck (StarMaze).
* **Transparent negatives.** Honest about failure modes (symmetries/partial observability; cue-centric tasks).

**Weaknesses:**

* **Per-layout pretraining assumption.** The hSFA extractor is pretrained separately for each environment layout. This is not an accepted assumption in standard embodied navigation and severely limits external validity. At minimum, show (i) cross-layout generalization (train on some layouts, test on unseen) and (ii) cross-environment/domain transfer (apply a pretrained extractor to totally different new environments) without re-pretraining.
* **Missing strong baselines (pose-interpretable).** Since the interpretability claim targets pose (location & heading), the most relevant comparators are explicit localization pipelines:

  * **(V)SLAM/VO → policy:** Feed oracle (or realistic noisy) pose ((x,y,\theta)) from a classical VSLAM/VO (e.g., ORB-SLAM/ORB-SLAM3) to the policy as a yardstick for “interpretable pose.”
  * **Neural-SLAM / mapping-augmented agents:** Lightweight learned mapper + policy to test whether hSFA’s gains come from having any localization/mapping signal vs. the specific SFA bias.
    If pretraining is allowed, also compare to navigation foundation/backbone encoders (large pretrained embodied vision models).
* **Limited robustness/generalization.** No tests for texture/lighting/layout changes, camera/FoV shifts, or sensor noise.

**Questions:**

1. **Per-layout pretraining:** Can you pretrain once across many layouts and evaluate on held-out layouts with zero re-pretraining?
2. **Cross-environment transfer:** Pretrain an hSFA extractor in Environment A and apply it to a totally different Environment B (different topology/appearance) without re-pretraining. How does PPO+hSFA perform vs. CNN/PCA and mapping/memory baselines?
3. **(V)SLAM/VO baseline:** Add a baseline based on a standard VSLAM/VO into the policy, to provide a clear "explicit" pos yardstick for your interpretability claim.
4. **Neural-SLAM / mapping baseline:** Add a learned mapping agent (e.g., lightweight Neural-SLAM/active mapping) to test whether improvements arise from having localization/mapping at all vs. the SFA bias.
5. **Navigation foundation/backbones:** If pretraining is allowed, compare against pretrained embodied encoders (navigation foundation models) to contextualize SFA’s benefits.
6. **Robustness:** Report results under texture/lighting/layout perturbations, camera FoV/noise, and mild domain shifts.

---

> ### Author Response · Authors · 2025-11-20
>
> Thank you for your valuable and thorough feedback, we have updated the paper accordingly and find it significantly improved. Changes are marked in red in the PDF. In particular:
>
> - We added an **oracle (ground truth)** and a **realistic noisy baseline** for comparison (Figure 5 and Section 5.2). Since they are non-visual, they are applicable only to StarMazeArm (where the goal position is fixed). We also added combined feature extractors using NatureCNN with ground truth or hSFA-derived pose information. Interestingly, ground truth (with or without noise) and **combined CNN/ground truth** perform worse than hSFA, suggesting hSFA’s slowness bias is indeed beneficial. We believe the reason is their decomposition of pose information in multiple decorrelated features of different granularity levels.
>
> - **SLAM** is now explicitly mentioned as related work in Section 2, previously it only appeared in the references. While we did not implement actual SLAM feature extractors, we believe the new baselines and combined extractors provide fair approximations of achievable SLAM performance in these environments, since SLAM likewise provides (noisy) location and heading information in addition to the visual observation perceived by the agent.
>
> - Regarding **robustness**, there is already comprehensive prior work, which we now refer to in Section 6: [1] shows lower-layer hSFA features are quite general and can thus be expected to be robust to perturbations such as lighting, and [2] finds that hSFA representations are relatively robust to certain changes in room layouts (such as morphing and scaling).
>
> - **Cross-layout** or **cross-domain generalization** is currently not possible with hSFA. We believe that no method would be able to extract pose when the agent is placed into an unknown room of which it only sees a small section. The currently available "offline" hSFA realization requires complete training (i.e. having seen the entire room) before it can extract location and heading from anywhere in the room. An online version (already discussed under limitations in the conclusion) could make it possible to learn pose information on the go while exploring an unknown room. In this present work, we aim to demonstrate the advantages and potential that offline hSFA already has, while encouraging future development of online variants.
>
> - Comparison to **navigation foundation models** is an interesting idea but we consider it beyond the scope of this discussion: Such models require vast training resources and amounts of data, while hSFA trains in minutes on a laptop.
>
> ----
>
> [1]: Pietro Berkes and Laurenz Wiskott. "Slow feature analysis yields a rich repertoire of complex cell properties." Journal of vision 5.6 (2005): 9-9.
>
> [2]: Fabian Schönfeld and Laurenz Wiskott. "Modeling place field activity with hierarchical slow feature analysis." Frontiers in computational neuroscience 9 (2015): 51.

---

### Official Review · Reviewer_Xut6 · 2025-11-02

**Soundness:** 3
**Presentation:** 3
**Contribution:** 3
**Rating:** 8
**Confidence:** 4

**Summary:**

Estimating the location and heading of an agent in an environment is an important question in Reinforcement Learning setups. In this manuscript, the authors argue that features learned using Slow Feature Analysis (SFA), a method from computational neuroscience, might provide a better alternative to tackle this problem. To demonstrate the feasibility of their approach, they first train hierarchical SFA networks, along with other control networks on randomly sampled 80,000 images from five different environments. They then train PPO agents including their feature extractors on various RL environments and demonstrate the benefits of their approach.

**Strengths:**

Overall, the manuscript was a delight to read--it is very well written and easy to read. The authors also provide the code to replicate their results, and discuss the limitations of their approach--a rare sight in AI these days. While the SFA idea itself might not be novel, its application to RL problems is interesting and provides significant insights to the field. I have a few suggestions which, I hope, help the authors improve the manuscript in weaknesess.

**Weaknesses:**

1 - The manuscript argues for the use of temporally smooth or "slow" features. Thus I believe an important comparison while arguing for its benefits would be to show the breakdown of the system when the features are not slow. I am unsure exactly how one would go about implementing a Fast Feature Analysis, but maybe by altering the estimation of the temporal derivatives? I imagine instead of doing $y(t+1) - y(t)$, the authors could try $y(t+n) - y(t)$ with an increasing $n$ (hopefully) breaking the performance?

2 - I find the justification of 80,000 images a little unsatisfactory, especially given the fact that the visual cue of the target ends up encoded. Maybe a better alternative could be to just show the performance of the models (on y-axis) across different dataset sizes (on x-axis). A plot like this will also help elucidate the sensitivity of the different methods used (maybe PCA fares better than hSFA with just 40,000 images).

3- Since performance often also depends on the parametric sizes, it would be good have a table listing those in the manuscript/Appendix.

4 - While not directly used for RL problems, I think there are also other efforts in AI that aim to implement similar constraints (in essence). The authors could have a look at and maybe better link their work to those of Contrastive Learning Through Time, or even models like V-JEPA, etc. I believe there are also efforts to link Object Permanence and Temporal smoothness in the literature.

Minor :
1 - There are a couple of typos in the article.

**Questions:**

I have written several suggestions in weakness section to hopefully improve the paper.
Overall, the manuscript offers a compelling approach to learn representations for RL problems and, with some revisions, could be well-positioned for publication.

---

> ### Author Response · Authors · 2025-11-20
>
> Thank you for the friendly and valuable feedback. It has helped us in improving our paper further, with changes marked in red in the PDF. Regarding your points:
>
> - **Fast SFA**: In practice, SFA is based on PCA of temporal derivatives. Here, those with the smallest variance (information that is natively provided by PCA) are the slowest. The easiest way to achieve fast feature analysis is to pick those with the largest variance. However, these will contain either noise or rapid oscillations, depending on the kind of environment. In either case, this information is of course not a good decision basis for agents. We agree that at a certain point, agent behaviour will degrade sharply, although we are not sure of the practical benefits of this result. Additionally, the threshold is likely quite environment-dependent. The time-lagged version of SFA that you have suggested here, by the way, is also an interesting technique. Its effect however is less to encourage fast features, but to ignore features that change with certain periodicities.
>
> - **Coverage**: We selected 80,000 hSFA samples because visual inspection showed that with fewer random steps, agents sometimes failed to reach parts of the environments. So we selected this deliberately high number to ensure maximally robust hSFA representations. In practice, far fewer samples may suffice, but this strongly depends on environment layout, so we do not think a detailed analysis would be generalizable. The key point is that the agent must have visited approximately all "views" in the environment to form accurate location and heading representations.
>
> - **Model parameter counts**: This is a good suggestion, we have added a table of parameter counts (Table 4 in the Appendix).
>
> - **Related literature**: The research directions you mention are interesting but since they do not directly relate to the RL setup we consider in this work, we have decided not to add them to the paper. However, another reviewer brought up SLAM, which is directly applicable to our setup and so we have now explicitly mentioned it under related work. Still, we think the works you mention are useful reading material for readers interested in getting a general picture of this field and we hope that readers will come across them here in the review section.
>
> - **Typos**: We went over the paper again and corrected any typos we found.

---

### Meta-Review · Area_Chair_v38E · 2026-01-04

**Summary:**

This submitted work addresses the problem of visual navigation in 3D environments learned end-to-end, in particular with RL. The work focuses on the estimation of location and heading and proposes slow features analysis instead of learning features entirely from the RL loss.

The paper has received four reviews with mixed ratings of 8, 2, 4 and 6. The most favorable reviewer appreciated a well written manuscript, and a discussion of limitations, but did not comment the positioning of the paper wrt to the actual problem it tries to solve - navigation.

The three other reviewers raised various issues but most importantly pointed to the very limited experimental validation in very limited experiments with limited physics and limited analyses.

The AC sides with the critical reviewers and would like to further extend these weaknesses:

- The experimental evaluation is sub par and in no terms comparable to the state of the art in visual navigation. Visual navigation in simulated environments is essentially a solved problem as far as the classical "PointGoal" setting is concerned even for far more realistic scenarios, where the environments corresponds to 3D scanned real buildings (eg. the Matterport scans of the 3 standard datasets of the domain, Gibson, Matterport3D and HM3D) and with continuous action spaces and actuation noise, and these settings are far more challenging than the environments used in this work.

- The choices of baselines are basically ablated version of the standard agent. No real navigation baseline has been used for comparison.

- This problem is related to visual SLAM and a positioning wrt to the SLAM community is missing.

- Beyond the comparisons with the simple baselines and the reconstruction of heading angles, there are only limited additional analysis or ablations.

The AC judges that in its current form the paper does not provide sufficient insights and that the claims of the paper need to be tested in more complex environments and compared to the state of the art in navigation.

**Reviewer Concerns:**

The main concerns mentioned above have not been addressed.

**Reviewer Scores:**

6pwf provided 4 and was very critical in their post-rebuttal communication, arguably more than in their initial review.

WM9V mentioned that they maintained 6.

---

### Decision · Program_Chairs · 2026-01-26

Reject